# Super Effective Removal of Toxic Metals Water Pollutants Using Multi Functionalized Polyacrylonitrile and Arabic Gum Grafts

**DOI:** 10.3390/polym11121938

**Published:** 2019-11-25

**Authors:** Ahmed M. Elbedwehy, Ali M. Abou-Elanwar, Abdelrahman O. Ezzat, Ayman M. Atta

**Affiliations:** 1Nanotechnology Center, Mansoura University, Mansoura 35516, Egypt; 2National Research Centre, Dokki, Giza 12622, Egypt; science_ali87@yahoo.com; 3Chemistry Department, College of Science, King Saud University, P.O. Box 2455, Riyadh 11451, Saudi Arabia; ao_ezzat@yahoo.com

**Keywords:** arabic gum, high efficient, acrylonitrile, superadsorbent, heavy metal

## Abstract

Super adsorbent polymers can be considered to be a very efficient solution for wastewater treatment. In general, their adsorption capacities depend on the type and amount of the functional groups present on the surface of the polymers, while their economic value is affected by their cost. Therefore, this study aims to understand the effect of multi-functionalization of cheap Arabic gum on the adsorption capability toward heavy metals. Graft copolymers of polyacrylonitrile (PAN) onto Arabic gum (AG) were prepared in aqueous solution using (KMnO_4_/HNO_3_) as a redox initiator. Chemical modification of the graft copolymer was carried out by reaction with hydrazine hydrochloride followed by hydrolysis in the basic medium. The modified graft product was characterized by various techniques, such as Fourier transform infrared spectroscopy (FTIR), elemental analysis, scanning electron microscope (SEM), and X-ray powder diffraction (XRD). The modified graft copolymer was used to adsorb Pb^2+^, Cd^2+^ and Cu^2+^ from their aqueous solutions using batch extraction. Different parameters influence the uptake behavior, including contact time, pH, and the initial concentration of the metal ions; all of these were investigated. The kinetics were investigated using the pseudo first order and pseudo second order, and the equilibrium data were analyzed using the Langmuir and Freundlich model. The modified graft product showed the superadsorbent capacity to obtain maximum values (Qmax) 1017, 413 and 396 mg/g for Pb^2+^, Cd^2+^ and Cu^2+^, respectively. Acid treatment with 0.2 M HNO_3_ resulted in 96%, 99% and 99% metal recovery for the Pb^2+^, Cd^2+^ and Cu^2+^, respectively. This indicates the recyclability of product for further usage upon drying between treatments.

## 1. Introduction

Water is the life artery of living systems, essential to human health and welfare, and a prerequisite to industrial development. Around 829,000 people are estimated to die each year as a result of unsafe drinking water [1]. Therefore, wastewater treatment is critical to the safety of human beings. The removal of toxic heavy metals such as lead, cadmium and copper is one of the main problems for wastewater treatment [2]. These metals can enter water systems through industrial mining, galvanoplastic foundries, and from pipe corrosion. Long-term exposure to these heavy metals can accumulate in living organisms, causing several disorders and serious diseases [2,3]. Therefore, it is necessary to search for economical and efficient methods to protect water resources from pollution. As a result of many efforts, precipitation, reverse osmosis, electrolysis and electrodialysis have been established for the recovery and removal of heavy metals from wastewaters [4]. However, these techniques are high in cost and may also be inefficient in the removal of some toxic metal ions, or otherwise produce toxic wastes of their own [5]. As a result, alternative inexpensive methods using naturally abundant resources are being investigated [6,7].

Among natural resources, polysaccharides have gained much attention due to their biodegradability, renewability and nontoxicity, making them an ideal candidate for use in water treatment [8]. Non crosslinked polysaccharides such as cellulose can be used for heavy metal adsorption or for making a crosslinking for soluble chitosan, sodium alginate, gum and others [9,10,11]. However, they suffer from some drawbacks such as fixed structure and limited functional groups, as well as low elasticity, which decreases its efficiency for heavy metal adsorption [12]. The grafting of synthetic polymers onto the natural polymers is one of the proposed techniques for modifying the chemical structures of polysaccharides and their metal adsorption capacities as bio-degradable non-toxic adsorbents. The metal adsorption capacity of starch [13] was enhanced by the grafting of itaconic acid onto its backbone [14]. The systematic grafting of polysaccharides has been carried out either with hydrophilic vinyl monomers or with polymeric hydrogels, and their use as effective adsorbents for the removal of heavy metals and organic water have previously been reported [15,16,17]. However, the modification of polysaccharides such as gums through graft copolymerization provides a tool in the hands of researchers for improving their properties [12]. Graft copolymerization of vinyl monomers such as acrylic acid, acrylamide, ethyl acrylate and their derivatives onto gums plays an important role in incorporating new chains with new properties into their backbone. Properties including elasticity, thermal stability and chemical reactivity based on new functional groups enhance their activity towards heavy metals [12]. Guar gum has been crosslinked and grafted with *N*,*N*-methylene-bis-acrylamide crosslinker [18] and poly methyl acrylate [19] to form a porous structure for trapping heavy metals from the aqueous solution. The formation of silica guar-gum-graft-polyacrylamide nanocomposites increased the efficiency of the gum adsorbent in removing cadmium metal ions, with a maximum adsorption capacity of 2000 mg/g [20]. Arabic gum (Acacia gum) composites were previously used to remove Cu^2+^ from their aqueous solution with a limited maximum adsorption capacity of 17.6–38.5 mg/g [21]. The present work aims to modify the chemical structure of Arabic gum with suitable functional groups using the grafting technique to enable their ability to form complexes with various metal ions. Acrylonitrile is one of most active monomers used for grafting [22]. Nitrile chains can be further modified by various other groups, such as modification with hydrazine, hydroxylamine and ethanolamine, to form amidrazone, amidoxime and oxazoline, respectively, and these groups are very active in removing toxic metals from wastewater [23,24,25]. In our study, we chemically modify polyacrylonitrile (PAN) graft Arabic gum by reaction with hydrazine, followed by basic hydrolysis of the remaining nitrile groups. The resultant product will have a superadsorbent character based on multi chelating groups such as amidrazone, carboxylate and amide groups.

## 2. Experimental

### 2.1. Materials

Acacia gum (BDH, England), acrylonitrile (AN), potassium permanganate and nitric acid (AR grade) were all purchased from (BDH, Poole, England). Hydrazine hydrochloride was supplied by Acros, Merelbeke, Belgium. Methanol and dimethylformamide (DMF) were purchased from Fisher, Indiana, USA. CuCl_2_.5H_2_O, Pb(NO_3_)_2_ and Cd(NO_3_)_2_ were used as sources for Cu^2+^, Pb^2+^ and Cd^2+^ ions, respectively, and were purchased from Sigma Aldrich, Missouri, MO, USA. Doubly distilled water was used throughout the experiment. Acacia gum was crystallized in distilled water followed by drying step before usage. All chemicals used in the experiment were of research grade.

### 2.2. Techniques

#### 2.2.1. Preparation of Polyacrylonitrile-Grafted Arabic Gum

The grafting reaction was carried out as previously described [25]. In a closed Pyrex cell, Arabic gum (0.5 g) was dissolved in a limited amount of distilled water to form a clear non-viscous solution. KMnO_4_ (0.033 mol/L) was freshly prepared in nitric acid (0.13 mol/L) solution and added to the gum solution. After that, AN (5 mol/L) was added dropwise under stirring. A polymerization experiment was carried out by placing the cell between a pair of tube fluorescent lambs (40 W) for 2 h. The polymerization process was terminated by adding methanol to the reaction mixture until full precipitation. The crude product was dried to constant weight in a vacuum oven at 40 °C. The homopolymer (polyacrylonitrile; PAN) was removed from the crude copolymer using Soxhlet extraction by DMF for 48 h. Finally, the obtained pure graft copolymer was dried under 40 °C until constant weight. The grafting percentage (GP = (*W*_g_ − *W*_p_) × 100/(*W*_p_); where *W*_g_ and *W*_p_ are the weights of pure graft and Arabic gum, respectively) was calculated and was found to be 193%. The GP value indicates that 193 units of PAN grafted onto 100 units of Arabic gum produce PAN–*g*–AG.

#### 2.2.2. Modification of the Graft Product with Hydrazine Hydrochloride

Five grams of Hydrazine hydrochloride in 25 mL of methanol and 2 g of PAN–*g*–AG were added to a reactor equipped with a magnetic stirrer and refluxed condenser. Then, a limited amount of 0.05 mol/L NaOH aqueous solution was added to the previous mixture until obtaining pH 9 [26]. The reaction was performed for 20 h at 80 °C under stirring. The modified product was washed thoroughly with water, and then with absolute methanol, followed by drying at 60 °C.

#### 2.2.3. Introduction of Carboxyl and Amide Groups to the Graft Product

The dried hydrazine-modified PAN–*g*–AG and 50 mL of 4% NaOH solution were added in a reaction vessel. Afterwards, the reaction mixture was heated to 75 °C for 6 h [27]. The modified product was precipitated in methanol and washed with water, diluted HCl (0.01 mol/L), until a neutral pH was obtained, and finally with methanol. The modified product was dried in an oven at 40 °C until constant weight.

### 2.3. Characterization of Samples

Atomic absorption spectrometer (Perkin-Elmer 500, Illinois, USA) was used to determine the metal ion contents in water by using copper, lead and cadmium cathode lamps. The apparatus was set at 283.30, 324.75 and 226.50 nm for lead, copper and cadmium, respectively. The surface morphologies of the samples were visualized using scanning electron microscopy (JEOL-JSM 5300, Tokyo, Japan), operating at a typical accelerating voltage of 20 kV. The samples were coated with gold before measurement. Infrared spectra (FTIR) were obtained by subjecting quantitative samples to IR spectrophotometer (Perkin Elmer 1430, Connecticut, USA). The products were subjected to X-ray diffraction (D8 Advance X-Ray Diffractometer; Bruker AXS, Wisconsin, USA) using Cu Kα radiation. The scattering angle 2θ was varied from 10° to 70°. Elemental analysis was done using a CHN analyzer from Perkin Elmer. Each sample was measured three times.

### 2.4. Metal Ion Uptake Experiments Using a Batch Method

Measurements of heavy metal ion uptake using a batch method were conducted by placing 0.2 g of modified PAN–*g*–AG in a beaker containing 500 mL (0.15 g/L) metal ion solution (Pb^2+^, Cd^2+^ and Cu^2+^) at pH 5.0. The contents of the beaker were shaken at 120 rpm at 28 °C. Samples were taken at time intervals of 2 min–24 h in order to evaluate the remaining metal concentration in solution. 

Heavy metal uptake experiments were conducted at controlled pH and 28 °C by shaking 0.05 g of dry modified PAN–*g*–AG with 50 mL (100 mg/L) metal ion solution (Pb^2+^, Cd^2+^ and Cu^2+^) for 4 h at 120 rpm. Five different pH concentrations from 1 to 6 were adjusted in suspensions of the polymer before adding the metal salt solutions. The pH of the medium was adjusted using buffer solutions of KCl/HCl for pH 1, 2, and 3; acetic acid/sodium acetate for pH 4 and 5; and Na_2_HPO_4_/KH_2_PO_4_ for pH 6. The ionic strength of the solutions was kept at 0.01 M for all of the different pHs in order to ignore the effect of ionic strength on the measurements. The influence of the initial concentration of the metal ions (Pb^2+^, Cd^2+^ and Cu^2+^) on the heavy metal uptake using modified PAN–*g*–AG was carried out by adding 0.05 g of dried product in a series of bottles containing 50 mL of metal ions at certain concentrations (200–1200 mg/L) and pH 5.0. The bottles were placed on a shaker at 120 rpm and 28 °C for 4 h to achieve chemical equilibrium. After adsorption, the remaining concentration of the metal ions was evaluated by withdrawing some solution using a syringe.

### 2.5. Desorption Experiments

Firstly, the modified PAN-g-AG was allowed to adsorb the maximum capacity of Pb^2+^, Cd^2+^ and Cu^2+^ metal ions before measuring the desorption %. Then, the PAN–*g*–AG samples were collected and washed vigorously with distilled water to remove any unadsorbed metal ions. Secondly, the washed PAN–*g*–AG was then shaken with 100 mL of nitric acid solution (0.05–0.2 mol/L). The remaining concentration of metal ions in the aqueous phase was evaluated by using an atomic absorption spectrophotometer (Perkin-Elmer, Shelton, CT, USA). The desorption efficiency is the percentage of metal ion concentrations in eluent divided by its initial concentration in the PAN–*g*–AG adsorbents. The reusability measurements were repeated for three consecutive cycles to investigate the reusability of PAN–*g*–AG. 

## 3. Results and Discussion

### 3.1. Preparation and Characterization of Modified PAN–g–AG

The proposed mechanism for the grafting of AN onto the starch by using KMnO_4_ was previously discussed in [28], which elucidated that the grafting was carried out by mean of a reduction radical mechanism. In this respect, the proposed chemical structure for preparing PAN–*g*–AG can be represented by Scheme 1. The mechanism is based on the reduction of Mn^7+^ to Mn^4+^ followed by subsequent reductions of Mn^4+^ to Mn^3+^ or Mn^2+^. The AG–CHO group produced from oxidation with KMnO_4_ was converted to enol form in order to produce a radical on the AG unit, while Mn^7+^ was reduced to Mn^4+^, Mn^3+^ or Mn^2+^ cations. The produced radicals of AG initiate the polymerization and grafting of AN onto AG. The two-step chemical modification of PAN–*g*–AG can be seen in Scheme 2 [29,30]. The chemical structures of PAN–*g*–AG and its modified structure with hydrazine and NaOH were elucidated from the FTIR spectra presented in Figure 1a–c. The IR spectrum of the PAN–*g*–AG (Figure 1a) and the hydrazine-modified PAN–*g*–AG (Figure 1b) show an absorption band at 2244 cm^−1^, confirming the presence of nitrile (CN) group. The appearance of two absorption bands at the 1667 and 1630 cm^−1^ regions (Figure 1a) is attributed to the stretching vibrations of C=N and N=N bands, respectively. The presence of the absorption band at 3360 cm^−1^ (attributed to the NH stretching vibrations of secondary amino groups) in Figure 1b, in addition to the N=N stretching vibration band at 1630 cm^−1^, confirms that the reaction of hydrazine with nitrile groups leads to the formation of amidrazones. However, cyclic compounds may be formed by a side reaction, as can appear in Scheme 3 [29,30]. Figure 1c confirms the formation of carboxamide and carboxylate functional groups (Scheme 2) based on the appearance of three absorption bands at 1408, 1559, and 1671 cm^−1^, which are attributed to C=O stretching in carboxamide functional groups, and symmetric and asymmetric stretching vibrations of carboxylate groups, respectively [29,30]. Furthermore, the shoulder beak at 3221 cm^−1^ is attributed to –NH stretching vibration. The disappearance of the nitrile band at 2244 cm^−1^ (Figure 1c) indicates that all cyanide groups are converted to carboxamide and carboxylate groups.

Another indication for the formation of cyclic nitrogen-containing structures can be deduced from the XRD data in (Figure 2a–c). The two X-ray diffractograms of PAN–*g*–AG (Figure 2a) and hydrazine modification of PAN–*g*–AG (Figure 2b) show two crystalline peaks at a maximum of nearly 2θ = 16° and 29°, which can be attributed to Brag planes (010) and (300), respectively. The crystalline structure of PAN–*g*–AG was expected, as a result of the special helical confirmation of PAN grafts due to the dipole interactions of nitrile groups [31]. The modification of PAN–*g*–AG with hydrazine (Figure 2b) shows peaks with lower intensity than that obtained for PAN–*g*–AG (Figure 2a) to confirm the greater ordering of nitrile dipole interaction with the hydrazine modification of PAN grafts [32]. On the other hand, Figure 2c reveals the amorphous nature of the basic modification of hydrazine PAN–*g*–AG may indicate the conversion of all nitrile groups to carboxamide and carboxylate during the progress of the saponification reaction.

The nitrogen contents of PAN-g-AG and its modified structure are determined and listed in Table 2 in order to determine the conversion increment of the nitrile [30]. The data indicate the increase in the nitrogen content for the modified hydrazine PAN–*g*–AG with different reaction times. Moreover, the relative nitrogen content of the hydrazine-modified copolymer (PAN–*g*–AG) decreased during the reaction from 18% to 4% (see Table 1).

The SEM images of PAN–*g*–AG copolymer and the modified PAN–*g*–AG are shown in Figure 3a–d. It can be seen that the PAN–*g*–AG (Figure 3a) has a tight and smooth surface, while the surfaces of the modified copolymer (Figure 3b–d) are porous and coarse, facilitating the adsorption of metal ions on the surface of the polymer [33].

### 3.2. Effect of Adsorption Conditions on Metal Ion Adsorption

The pH of the metal ion solution plays a major role in the whole adsorption process, especially on adsorption capacity, by modifying the level of ionization of superadsorbent-modified PAN–*g*–AG [34]. To investigate the effect of different pH values on the metal uptake by the modified PAN–*g*–AG, five different pH concentrations from 1 to 6 were adjusted in suspensions of the polymer before adding the metal salt solutions. This pH range was chosen to avoid the precipitation of metals as metal hydroxide as a result of excessive pH, and hence the removal of the metal could be dependent on the adsorption process only [35]. In Figure 4, the data show that the amount of metal ions adsorbed by modified PAN–*g*–AG gradually increased from pH 1 to 6, reaching a peak at pH 5 in the case of Pb^2+^ and at pH 6 in the case of Cd^2+^ and Cu^2+^. The specific capacities decreased in the following sequence: Pb^2+^ > Cd^2+^ > Cu^2+^. The modified PAN–*g*–AG showed high efficiency in removing heavy metals, even at pH 2, where about 45%, 37% and 25% of Pb^2+^, Cd^2+^ and Cu^2+^ at an initial concentration of 100 mg/L of metal ions could be absorbed. This attribute may be because a result of the synthesized modified PAN–*g*–AG having the advantage of combining various functional groups with different exchange characteristics, such as 4-amino-l,2,4-triazole, dihydrotetrazine and tetrazine groups, as well as carboxylate and carboxamide functions. Therefore, the ion exchange seems to meet one of the highly efficient polymers for the removal of heavy metal ions from acidic and neutral water.

The effect of the initial concentration of heavy metal ions on the adsorption aptitudes of Pb^2+^, Cu^2+^ and Cd^2+^ on modified PAN–*g*–AG was determined over 24 h, and the data are presented in Figure 5. It is clearly apparent that the adsorption abilities increased with the increase in the initial concentration of heavy metal ions, particularly for Pb^2+^. The maximum amounts of adsorption for Pb^2+^, Cd^2+^ and Cu^2+^ were 1017, 413 and 396 mg/g, respectively, while the capacity of adsorption decreased in the following sequence: Pb^2+^ >> Cd^2+^> Cu^2+^ under the same conditions. This attribute may be explained by the fact that adsorption capacity was mainly related to the ionic charges, hydrated ionic radius and the outer electron configurations of the metal ions [36]. Pb^2+^, Cd^2+^ and Cu^2+^ have the same ionic charge; however, the valence electron configuration of Pb^2+^ was 6s^2^, while for Cd^2+^ and Cu^2+^ it was 3d^8^ and 3d^9^, respectively. Furthermore, Pb^2+^ has a hydrated ionic radius larger than that of Cd^2+^ and Cu^2+^. Consequently, it was easier for hydrated Pb^2+^ to lose the coordinated water and form a stable valence electron configuration of 6s^2^ when reacted with the adsorption sites of modified PAN-g-AG. Therefore, the adsorption capacity of Pb^2+^ was much higher than that of Cd^2+^ and Cu^2+^.

The effect of contact time was studied by varying the equilibrium time from 2 min to 24 h at pH 5 for Pb^2+^, Cd^2+^ and Cu^2+^ adsorption onto modified PAN–*g*–AG. Figure 6 clearly demonstrates that the amount of Pb^2+^, Cd^2+^ and Cu^2+^ adsorbed onto modified PAN–*g*–AG was nearly 93% after only 2 min for an initial concentration metal ion of 150 mg/g. Subsequently, the adsorption rate rises gradually and reaches equilibrium after 2.5 h. The short time required to reach equilibrium revealed that modified PAN–*g*–AG has a very high adsorption efficiency and a great potential in Pb^2+^, Cd^2+^ and Cu^2+^ adsorbent applications [37].

Comparing the adsorption capacity data of the base modified PAN–*g*–AG of 1017, 413 and 396 mg/g for Pb^2+^, Cd^2+^ and Cu^2+^, respectively, with the previously reported data for the modified PAN fibers [29], higher adsorption values are revealed for the base modified PAN–*g*–AG. The PAN fibers achieved 371.42, 400.4, and 392 mg/g for Pb^2+^, Cd^2+^ and Cu^2+^, respectively. These data indicate the higher ability of the chemically grafted biopolymer PAN–*g*–AG and very low cost superadsorbent material to desorb toxic metal ions from aqueous solutions.

### 3.3. Equilibrium Adsorption Studies

Equilibrium adsorption isotherms have a chief role in the study and design of adsorption systems [38]. The adsorption of metal ions onto the surface of solid substances may be explained by various isotherms, including Langmuir, Freundlich, Sips and Redlich-Peterson isotherm equations [38]. The Langmuir and Freundlich isotherms express the coverage or adsorption of molecules on a solid surface and the concentration of a medium above the solid surface at a defined temperature. In this study, the analysis of experimental data was carried out using both the Langmuir and Freundlich isotherm models. The linear form of the Langmuir isotherm equation is given as:(1)1qe=1bqm+1Ce
where *b* is the Langmuir equilibrium constant (l/mg), and qm (mg/g) is the monolayer adsorption capacity. Both are calculated from a plot *C_e_*/*q_e_* versus *C*_e_ (Figure 7a). The Langmuir parameters were determined from the linear relations (Figure 7a) and are listed in Table 2. The data show that the adsorptions of all metal ions have higher linear coefficients (*R*^2^) when they are nearer to 1. This elucidates the homogeneity of PAN-g-AG surfaces (Figure 3b–d) and the formation of the monolayer of the adsorbed metal ions onto the modified PAN–*g*–AG surfaces. Moreover, the agreement between the values of the experimental adsorption capacities (*Q_exp._*) and those calculated (*Q*_m_) using the Langmuir model confirms that the adsorption process of all metal ions for Pb(II), Cd(II) and Cu(II) obey the Langmuir model. 

The Langmuir isotherm is evaluated by a separation factor, *R_L_*, which is defined as follows:(2)RL=11+bC0
where *C*_0_ in this case is the highest initial solute concentration. The value of separation factor indicates the isotherm type and the nature of the adsorption process. According to the *R_L_* value, the adsorption can be unfavorable (*R_L_* > 1), linear (*R_L_* = 1), favorable (0 < *R_L_* < 1) [39] or irreversible (*R_L_* = 0). In our study, all the *R_L_* values were found to be (0 < *R_L_* < 1), which demonstrates that the superadsorbent polymer shows favorable adsorption for Pb(II), Cd(II) and Cu(II) ions. The *R_L_* values indicate the favorability for adsorption of Pb^2+^ on modified PAN–*g*–GA (*R_L_* = 0.058) > Cd^2+^ (*R_L_* = 0.014) > Cu^2+^ (*R_L_* = 0.008).

The Freundlich isotherm is purely empirical, and it is the best model for explaining the adsorption on heterogeneous surfaces [40]. The Freundlich isotherm equation can be calculated in its linear form as follows:(3)qe=KfC1/ne
where *K_f_* (l/g) is the Freundlich constant and n is the Freundlich exponent. These parameters were calculated from a plot log *q_e_* versus log *C_e_* (Figure 7b) and summarized in Table 2. The magnitude of 1/*n* measures the favorability of adsorption and the degree of heterogeneity of the superadsorbent polymer surface. If 1/*n* is less than 1, this indicates a favorable adsorption, so adsorption capacity increases and new adsorption sites form. The values of 1/*n* indicate the favorability of adsorption on the superadsorbent polymer surface.

### 3.4. Adsorption Kinetics

To clarify the adsorption kinetics process of Pb^2+^, Cd^2+^ and Cu^2+^ on the modified PAN–*g*–AG, two kinetic models, a pseudo-first-order model and a pseudo-second-order model, were applied to the experimental data. Pseudo-first-order models are always used when studying kinetics [41]. This is explained by the following equation:(4)log(qe− qt) =logqe−K12.303t
where *q_e_* and *q_t_* (mg/g) are the quantity of metal ions (II) adsorbed at equilibrium and at time *t*, respectively, whereas *k* is the equilibrium constant (min^−1^), which was obtained from the slopes of the linear plots of ln (*q_e_* − *q_t_*) versus *t* (Figure 8a). The pseudo-second-order model can be expressed by the following equation:(5)tqt=1K2q2e+1qet
where *K*_2_ (g/mg min) is the equilibrium rate constant for the pseudo second-order adsorption, whereas *q_e_* can be calculated from the plot of *t*/*q_t_* against *t* (Figure 8b) [42]. A comparison of the results with the correlation coefficients for the first and second order models is listed in Table 3. The data indicates that the pseudo-second-order mode with higher *R*^2^ value have the best fit model for the experimental kinetic data.

Intraparticle diffusion equation was also investigated, as Equations (4) and (5) cannot identify the diffusion mechanisms [43,44]. The intraparticle diffusion model is obtained from the following equation:(6)qt=kintt0.5+C
where *k_int_* (g/mg min^1/2^) is the constant of the adsorption, *C* is the intercept and can be calculated from a plot *q_t_* versus *t*^1/2^ (Figure 8c) [45]. These plots may reveal a multi-linearity [46,47], indicating that two or more steps take place. The first, sharper portion is the external surface adsorption or instantaneous adsorption stage. The second portion is the gradual adsorption stage, where intraparticle diffusion is rate-controlled and *k_int_* is obtained from it. The third portion is the final equilibrium stage, where the adsorbate has extremely low concentrations in the solution due to the slowing down of intraparticle diffusion.

The kinetic expression given by Boyd et al. was further used for analyzing the kinetic data [48]. To determine whether adsorption proceeds with an external diffusion or intraparticle diffusion mechanism is given as follows:(7)F(t)=1−6π2∑n=1∞1n2exp(−n2BT)where *F* is the fractional of equilibrium at different times (*t*), and *B* (*t*) is a mathematical function of *F*, *n* is an integer that explains the infinite series solution and *F* is the fractional attainment of equilibrium at time *t*, which is obtained by the following:(8)F=qtqewhere *q_t_* and *q_e_* are the amount of metal ions adsorbed at the time (*t*) and equilibrium respectively. Reichenberg [49] obtained the following approximations:

For *f* values > 0.85;
(9)B(t)=−0.4977−ln(1−f)

For *f* values < 0.85;
(10)B(t)=[π−π−(π2F(t)3)]2

The data listed in Figure 8c confirm that none of the intraparticle diffusion plots passed through the original point, demonstrating that the intraparticle diffusion was part of the adsorption. Moreover, it was not the only rate-controlling step, which indicates the effect of film diffusion (boundary layer diffusion) on adsorption of lead, copper and cadmium. Also, the Boyd plots (Figure 8d) did not pass through the original point, which indicates that film diffusion is the rate-limiting adsorption process for the three heavy metals onto the surface of the polymer.

### 3.5. Desorption Characteristics

The desorption data of the different metal ions from PAN–*g*–AG adsorbents using 0.2 M HNO_3_ are summarized in Table 4, and show that the desorption efficiency of Pb^2+^, Cd^2+^ and Cu^2+^ were 96%, 99% and 99%, respectively. Moreover, the desorption efficiencies increase with increasing HNO_3_ concentration from 0.05 to 0.2 M, while moderate desorption efficiencies of metal ion (84%) occurred at 0.1 M HNO_3_. Therefore, the best concentration of HNO_3_ was determined to be at an economic concentration of 0.1 M, even after five desorption cycles. The adsorption capacity of the modified PAN–*g*–AG could still be retained at the 99% level at the 3rd cycle, as shown in Table 5. These results revealed that there was no appreciable loss in activity over at least three cycles.

## 4. Conclusions

The chemically modified graft product PAN–*g*–AG was obtained by two step synthesis to make carboxylate and carboxamide groups containing hydrazine. The modified product showed highly efficient uptakes of 1017, 413 and 396 mg/g for Pb^2+^, Cd^2+^ and Cu^2+^, respectively. The adsorption kinetics followed the mechanism of the pseudo-second-order equation for all systems studied, indicating chemical sorption to be the rate-limiting step of the adsorption mechanism. The metal ion-loaded modified PAN–*g*–AG was regenerated with an efficiency of 65%–99% using 0.05–0.2 M HNO_3_. Based on previous results, it can be concluded that the ability to chemically modify a graft biopolymer containing nitrile groups for use as a very low cost and superadsorbent material for heavy metal ions was effectively achieved.

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
