# Peer review of "Super Effective Removal of Toxic Metals Water Pollutants Using Multi Functionalized Polyacrylonitrile and Arabic Gum Grafts"

_polymers, 2019, doi:10.3390/polym11121938_

Round 1
Reviewer 1 Report
Derivatized gums by adding functional groups to efficiently remove toxic metals is a matter of great interest. Graft copolymers have been tested in the recent past. Although the manuscript is not a review, I am missing some more literature data either in the introduction or in the discussion section: “super” effectiveness must be proven by comparing these results versus some other sorbents, especially those involving gums and PAN.
With regard to the details, I would like to indicate some points that should be corrected prior to publication.
Line 86: What is the basis of calculation for the GP? Is the obtained value abnormally high, or a typical one for these reactions?
Lines 120-121: What are the ionic strengths of the buffer solutions? Do they have an effect on the adsorption? How can this be proved/tested?
Lines 144,145,147: “is probably due”, “may be due to” do not point towards “these data confirm”. This discussion paragraph should be checked and improved.
Line 160: The reported increase is too small to be significant. What do the authors mean by “a significant decrease in the total amorphous area” (I guess ref. 22 is incorrect here, see below).
Scheme 1: I am afraid I cannot follow the grafting procedure (lines 76-86) by looking at this scheme. Could you please improve this point?
Pages 8, 9, table 2: I am missing some further discussion regarding the comparison between the Langmuir and Freundlich models results.
There is a problem with the references. I did not have time to check all of them, but I think that, on line 140, it should say ref. 19 & 20, so there is some wrong numbering somewhere.
English spelling and grammar should be checked throughout the whole text. See, for instance, lines 35, 36, 42, 81, 106, 115, 153, 195, 204, 254, …
Author Response
Comments and Suggestions for Authors
Graft copolymers have been tested in the recent past. Although the manuscript is not a review, I am missing some more literature data either in the introduction or in the discussion section: “super” effectiveness must be proven by comparing these results versus some other sorbents, especially those involving gums and PAN.
Answer: new paragraphs added in the introduction section and discussion for comparison.
Line 86: What is the basis of calculation for the GP? Is the obtained value abnormally high, or a typical one for these reactions?
Answer: The basis of calculation was clarified in the text line 100-104.
Lines 120-121: What are the ionic strengths of the buffer solutions? Do they have an effect on the adsorption? How can this be proved/tested?
Answer: The ionic strength of the solutions was kept at 0.01M for all different pHs to ignore the effect of ionic strength on the measurements.
Lines 144,145,147: “is probably due”, “may be due to” do not point towards “these data confirm”. This discussion paragraph should be checked and improved.
Answer: The section is improved
Line 160: The reported increase is too small to be significant. What do the authors mean by “a significant decrease in the total amorphous area” (I guess ref. 22 is incorrect here, see below).
Answer: references modified with the new references 31 and 32.
Scheme 1: I am afraid I cannot follow the grafting procedure (lines 76-86) by looking at this scheme. Could you please improve this point?
Answer: Scheme 1 added to illustrate this point
Pages 8, 9, table 2: I am missing some further discussion regarding the comparison between the Langmuir and Freundlich models results.
Answer:New pargraph added to clarify the data
There is a problem with the references. I did not have time to check all of them, but I think that, on line 140, it should say ref. 19 & 20, so there is some wrong numbering somewhere.
Answer: The references corrected and modified.
English spelling and grammar should be checked throughout the whole text. See, for instance, lines 35, 36, 42, 81, 106, 115, 153, 195, 204, 254, …
Answer: The article revised and corrected
Reviewer 2 Report
1)In the introduction section, others grafted polysaccharides have been employed as metal removed, for example: Journal of Polymers and the Environment, 24(4), pp. 343-35
2) An additional scheme should be included related the preparation of Polyacrylonitrile grafted Arabic gum.
3) Please explain 193% grafting percentage.
4) I do not understand Figure 6. In this Figure, the Qe reach values of aprox 370 mg/g for Pb in the same range as for Cd and Cu. However, in other experiment the adsorption of Pb is always higher. In fact, in Figure 5, the adsorption of Pb for initial metal ions concentration 150 mg/l (240 min) reach 800-900 mg/g, whereas in Figure 6 for the same initial concentration, time (240 min) and pH (5) the adsorption only reach values of 370 mg/g.
5) In the desorption study, the authors said that the best concentration of HNO3 was M as an economic concentration, however desorption only reaches about 84-90%. From which sample repeated adsorption of Pb2+, Cu2+, and Cd2+ ions were done?
6)The English of the manuscript should be polished. There are some grammar mistakes along the text, for instance:
Line 24: were instead was
Line 59: other groups
7) Also, there the manuscript should be carefully revised as there are other errors and aspect to improve:
Few examples:
Line 68, remove parenthesis
Line 88: Hydrazine in lower case
Line 97: HCl
Line 104: “were visualized using scanning electron microscopy”
Celsius degree is written in different ways along the manuscript.
Line 157, 158: removed spectrum, they are not spectra, they are diffractograms.
Lines 294-296 different size.
The quality of all figures, tables and equations should be improved. The equations and figures seems to be stretched. Table 1 has part of the cells empty.
Caption of Figure 3, “Modified PAN-g-AG superadsorbent at different magnification scale”, Which sample? Hydrazine modified or Base modification of hydrazine-modified? This explanation should be added.
Author Response
Reviewer 2
In the introduction section, others grafted polysaccharides have been employed as metal removed, for example: Journal of Polymers and the Environment, 24(4), pp. 343-35
Answer: The introduction section modified and marked with the red color
An additional scheme should be included related the preparation of Polyacrylonitrile grafted Arabic gum.
Answer: New scheme added.
3) Please explain 193% grafting percentage.
Answer: The basis of calculation was clarified in the text line 100-104.
I do not understand Figure 6. In this Figure, the Qe reach values of aprox 370 mg/g for Pb in the same range as for Cd and Cu. However, in other experiment the adsorption of Pb is always higher. In fact, in Figure 5, the adsorption of Pb for initial metal ions concentration 150 mg/l (240 min) reach 800-900 mg/g, whereas in Figure 6 for the same initial concentration, time (240 min) and pH (5) the adsorption only reach values of 370 mg/g.
Answer: Figure 6 show the effect of contact time on the adsorption of metal ion at definite pH 5 and 150 mg/L. While Figure 5 shows the effect of metal ion concentration on the Qe at various time intervals. So it cannot compare between two figures because the contact time was same for both figures 5 and 6. The effect of initial concentration of heavy metal ions on the adsorption aptitudes of Pb2+, Cu2+ and Cd2+ on modified PAN-g-AG was carried out for 24h and the data were represented in figure 5.
In the desorption study, the authors said that the best concentration of HNO3 was M as an economic concentration, however desorption only reaches about 84-90%. From which sample repeated adsorption of Pb2+, Cu2+, and Cd2+ ions were done?
Answer: Therefore, the best concentration of HNO3 was determined to be 0.1 M as an economic concentration even after five desorption cycles
6)The English of the manuscript should be polished. There are some grammar mistakes along the text, for instance:
Line 24: were instead was
Line 59: other groups
7) Also, there the manuscript should be carefully revised as there are other errors and aspect to improve:
Few examples:
Line 68, remove parenthesis
Line 88: Hydrazine in lower case
Line 97: HCl
Line 104: “were visualized using scanning electron microscopy”
Celsius degree is written in different ways along the manuscript.
Line 157, 158: removed spectrum, they are not spectra, they are diffractograms.
Lines 294-296 different size.
The quality of all figures, tables and equations should be improved. The equations and figures seems to be stretched. Table 1 has part of the cells empty.
Answer: The article revised and corrected
Caption of Figure 3, “Modified PAN-g-AG superadsorbent at different magnification scale”, Which sample? Hydrazine modified or Base modification of hydrazine-modified? This explanation should be added.
Answer: (b), (c) and (d) base modified PAN-g-AG superadsorbent at different magnification scale.
Round 2
Reviewer 1 Report
In my opinion, all the previous reviewers' concerns were properly addressed, so I recommend publication.
Please check the sentence: "The mechanism is based on the oxidation of Mn7+ to Mn4+ followed by reduction of Mn4+ to Mn3+ or Mn2+."
Author Response
Please check the sentence: "The mechanism is based on the oxidation of Mn7+ to Mn4+ followed by reduction of Mn4+ to Mn3+ or Mn2+."
Answer: the sentences converted to The mechanism is based on the reduction of Mn7+ to Mn4+ followed by reduction of Mn4+ to Mn3+ or Mn2+. The AG-CHO group produced from oxidation with KMnO4 converted to enol form to produce radical on the AG unit and Mn7+ reduced to Mn4 .
Reviewer 2 Report
The manuscript improves significantly, but I still have some concern about Figure 5 and 6.
I agree with the authors that the Figures showed different effect, but some points of the representations should be the same.
In figure 5, for an initial concentration of Ce= 150 mg/g at pH 5 (as it is described in experimental section) and 24 h, the Qe value for Pb2+ is around 800 mg/g. In contrast in Figure 6, again for an initial concentration of Ce= 150 mg/g at pH 5 and 24 h (1440 min in the graph) the Qe value for Pb2+ is around 370 mg/g. Why such difference?
Although the Figures display different effects, these two points in the graphs should be the same because the experiments were done at the same conditions
Author Response
Comments and Suggestions for Authors
The manuscript improves significantly, but I still have some concern about Figure 5 and 6.
I agree with the authors that the Figures showed different effect, but some points of the representations should be the same.
In figure 5, for an initial concentration of Ce= 150 mg/g at pH 5 (as it is described in experimental section) and 24 h, the Qe value for Pb2+ is around 800 mg/g. In contrast in Figure 6, again for an initial concentration of Ce= 150 mg/g at pH 5 and 24 h (1440 min in the graph) the Qe value for Pb2+ is around 370 mg/g. Why such difference?
Although the Figures display different effects, these two points in the graphs should be the same because the experiments were done at the same conditions
Answer: in the experimental part line 132 is corrected to (0.2 g of modified PAN-g-AG in a beaker having 500 mL (0.15 g/L) metal ion solution (Pb2+, Cd2+).
In figure 5, the relation between qe (maximum adsorption capacity) and Ce (the remaining metal ion concentrations after 24h adsorption) we plotted using different C0 (initial metal ion concentrations) as mentioned in the experimental part (a series of bottles containing 50 mL of metal ions at certain concentrations (200–1200 mg/l) and pH 5.0.). the values of qe in figure 5 was too high for the Pb2+ uptake because the high initial concentrations (Co) we used.
In figure 6 for studying the kinetics of metal ions removal we unify the initial concentration (Co) for all the metal ions to be 150 ppm as mentioned in the experimental part (0.2 g of modified PAN-g-AG in a beaker having 500 mL (0.15 g/L) metal ion solution (Pb2+, Cd2+ and Cu2+) at pH 5.0). Due to that the maximum uptake in figure 6 for Pb2+ only reach qe=375 mg/g. If we used higher initial concentration (Co) in figure 6, the value would be fixed for (Cd2+ and Cu2+) while it would be high for Pb2+ same as it appeared in figue 5. That is the reason of the difference between the qe values fo Pb2+ in figures 5,6. Furthermore, the value of 150 ppm in the x- axis in figue 5 is related to Ce value while the value 150 ppm in the figure 6 is related to Co.
Round 3
Reviewer 2 Report
accept in present form